# Crop-Livestock Integration Improves Physical Soil, Agronomic and Environmental Aspects in Soybean Cultivation

**DOI:** 10.3390/plants12213746

**Published:** 2023-11-01

**Authors:** Jordaanny Danyelly Pereira Lima, Aline Borges Torino, Luciana Maria da Silva, Lucas Freitas do Nascimento Júnior, Marlete Ferreira de Brito, Kátia Aparecida de Pinho Costa, Bruno Montoani Silva, Eduardo da Costa Severiano

**Affiliations:** 1Graduate Program in Agricultural Sciences/Agronomy, Instituto Federal Goiano, Rio Verde 75901-970, Brazil; jordana-17@hotmail.com (J.D.P.L.); alinetorino@hotmail.com (A.B.T.); luy.mari@hotmail.com (L.M.d.S.); lucasfnj@hotmail.com (L.F.d.N.J.); marleterv@gmail.com (M.F.d.B.); katia.costa@ifgoiano.edu.br (K.A.d.P.C.); 2Departament of Soil Science, Universidade Federal de Lavras, Lavras 37200-000, Brazil; brunom.silva@ufla.br

**Keywords:** conservation agriculture, *Brachiaria* sp., least limiting water range, biological soil loosening, provision of environmental services

## Abstract

Soybean is one of the most widely grown crops in the world and technologies are increasingly needed to increase productivity without impacting environmental degradation. In this context, the aim was to evaluate the action of forage plants of the genus *Brachiaria* sp. in crop–livestock integration on physical soil, agronomic and environmental aspects of soybean cultivation. The experiment was conducted in a subdivided plot design with seven integrated systems corresponding to the previous cultivation of Paiaguas palisadegrass, Xaraes palisadegrass and Ruziziensis grass in monocropping and intercropped with maize, as well as maize in monocropping. In the subplots, two grass management systems were evaluated: free growth and a grazing simulation cut. The bulk density and least limiting water range were assessed using soil samples and, after the pastures were desiccated when the soybean crop was planted, straw decomposition and plantability. A soil physics diagnosis by the bulk density and least limiting water range showed that the Paiaguas palisadegrass and Xaraes palisadegrass improved the soil environment due to biological soil loosening. The remaining mulch biomass did not affect soybean sowing and the adoption of *Brachiaria* sp. grass in the off-season, in addition to contributing to the provision of environmental services, and did not compromise grain productivity in succession.

## 1. Introduction

In recent decades, Brazil has been a world leader, especially in agribusiness [1,2], due to technological innovations, the efficient use of inputs and large-scale cultivation resulting from agricultural mechanization. It is a country that has the necessary resources to increase food production and contribute to the world’s food supply [3,4]. The use of this technology has made it possible to expand the frontier to previously marginalized regions, especially the Cerrado region, which has contributed greatly to increasing the country’s grain, livestock and agro-energy production [5,6].

In this context of growing agricultural production, the risks to the conservation of natural resources and the importance of the sustainability of the production system have been highlighted [7,8]. Conservation management systems contribute to maintaining or improving soil health [9,10] in order to provide suitable conditions for crop development with a consequent increase in productivity. However, these aspects have been neglected by some farmers, inducing soil compaction and triggering consequences such as reduced productive capacity [11,12], increased soil degradation and erosion processes [13,14].

Compaction is a consequence of the compression of unsaturated soil when subjected to the pressures applied by machinery and animals [15,16] and is characterized as the greatest physical limitation to high crop productivity worldwide [17,18]. For this reason, the recovery of the soil’s physical quality is of agricultural and environmental importance, especially in tropical regions, such as the Latossolos in Cerrrado [in the Brazilian Soil Classification System—Santos et al. [19]; Oxisols in the USA Keys of Soil Taxonomy—Soil Survey Staff (2014); and Ferralsols in the World Reference Base for Soil Resources—IUSS Working Group WRB (2015)], in soils that are highly susceptible to compaction [13]. However, due to the high demand for investments in the mechanical decompaction process, alternative techniques have been sought to amortize operating costs and ensure agronomic sustainability [4,20].

Crop–livestock integration (CLI), when associated with the cultivation of *Brachiaria brizantha* (syn. *Urochloa*), is a promising technique for biological soil loosening and improving the structural quality of the soil [16,21,22,23], particularly in intensive agricultural production systems where machine traffic is intense [24]. Thus, the development of these plants can mitigate the deleterious effects caused to the soil structure [11]. As a result, there is an increase in water infiltration in the soil and subsequent availability for crops, as well as a reduction in surface runoff, which prevents the erosion and pollution of water sources [25,26], contributing to the maintenance of environmental services. An increase in agricultural productivity has also been reported with a reduction in the use of inputs, providing an agronomically efficient and environmentally friendly production system [4,27,28].

However, the varietal and agronomic management of *Brachiaria* sp. is a determining factor in the success of farming activities, without the counterpart of environmental degradation. In this context, the aim was to evaluate aspects of conservation agriculture in the development of the soybean in the crop–livestock integration system, under the remaining mulch biomass of forage species of the *Brachiaria* genus in the Latossolo Vermelho Acriférrico in the Brazilian Cerrado.

## 2. Results and Discussion

### 2.1. Physical Quality of the Soil

The relationship between the soil water content and bulk density (Bd), considering the critical limits of the least limiting water range (LLWR), is shown in Figure 1, where the permanent wilting point (θ_PWP_) limited the LLWR up to 1.24 kg dm^−3^. From this point, θ_PR_ becomes the lower limit of the LLWR, whose amplitude decreases until nullity at Bd = 1.36 kg dm^−3^, considered the critical density (cBd). Similar results are reported by Flávio Neto et al. [21], both in oxide and clayey Latossolos which show no aeration limitation. However, the critical soil penetration resistance is reached at the same statistical point in the soil structure (Bd = 1.25 kg dm^−3^) as that observed by Severiano et al. [29] in a Latossolo from the same region and with a similar texture.

The soil water content at which the minimum air porosity is reached (θ_AP_) decreases with increasing Bd, without, however, limiting the magnitude of the LLWR (Figure 1). This is due to the development of large proportions of structural pores in oxide Latossolo [30], which in turn favor excessive aeration. The authors attribute this behavior to the granular structure of Latossolo, which causes low capillarity and, consequently, lower water retention [29]. Soil penetration resistance increased the most with soil compaction, which results in an increase in the mechanical resistance of the soil, hindering the growth and development of the plant root system.

When analyzing the LLWR, it can be seen that there is an increase in the soil water content up to the bulk density value of 1.24 kg dm^−3^. This value is, therefore, considered beneficial (bBd), indicating that a small increase in the Bd can increase water retention in these soils, since they are highly porous soils under natural conditions [29]. Above the bBd, the magnitude of LLWR decreases until the cBd is at 1.36 kg dm^−3^, where LLWR = 0. Under these conditions, plant development is severely limited by the physical degradation of the soil [31] (Figure 1). The range of LLWR varied between 0.0 and 0.14 dm^3^ dm^−3^, values which are common in Latossolo of the same textural class and subjected to intensive management [12,21,32].

Figure 2 shows the soil’s physical quality attributes as a result of the integrated systems adopted. It should be noted that the grazing simulation cut of the forage plant did not differ statistically from the grass in free growth.

In this study, it was expected that the grazing simulation would promote greater structural changes in the soil compared to grass in free growth. This is because cutting forage plants induces the emission of new tillers and increases the root density in grazed areas [6], thus intensifying soil structuring. However, the grasses were grown during the Brazilian winter period on the occasion of crop–livestock integration [32], a time characterized by low rainfall which affected their full development [33]. On the other hand, Flávio Neto et al. [21], when assessing the potential for biological soil loosening by grasses of the *Brachiaria* sp. genus, found that the resumption of summer rains was the factor responsible for effective root action on the soil structure. It is also suggested that in these soil and climate conditions, the rooting responses of *Brachiaria* grasses and, consequently, the structural changes in the soil are associated with the genotype as opposed to the way the plant is used.

It can be seen that the Bd values exceeded the bBd (1.24 kg dm^−3^; Figure 1 and Figure 2), indicating that the soil has physical quality restrictions. On the other hand, there was a differentiated effect of forage systems on the soil structure through Bd (Figure 2A), with a higher efficiency of biological loosening in *Brachiaria brizantha* forages (Xaraes and Paiaguas), even with a small variation in the amplitude of this attribute (1.26 to 1.34 kg dm^−3^), in the following order: maize in monocropping ≥ Ruziziensis grass in monocropping = Ruziziensis grass intercropped ≥ Paiaguas palisadegrass intercropped > Paiaguas in monocropping = Xaraes in monocropping = Xaraes palisadegrass intercropped with maize. Due to the multifactorial nature of LLWR, it has been considered the biophysical soil indicator that best correlates with plant growth [12,22], being that Figure 2B shows an amplitude of 3.5 times (0.04 to 0.14 dm^3^ dm^−3^) and demonstrates its high sensitivity to structural changes in the soil of these *Brachiaria* plants.

Among the forage systems adopted, *Brachiaria brizantha* (Xaraes and Paiaguas) stands out as a strategy for mitigating physical soil degradation, with the highest LLWR and Bd values close to the bBd. This result is due to the intense emission of tillers and roots [28] and, in the case of the Xaraes cultivar, also due to the speed of regrowth and forage production [23]. Perhaps this is why this grass can provide the environmental service of biological soil loosening, even when intercropped with maize. Similar results were found by Flávio Neto et al. [21] when evaluating LLWR as a result of the species *B. brizantha* cv. Xaraes and Piatã, *B. decumbes* and *B. ruziziensis* during the winter period.

On the other hand, the treatments with maize in monocropping showed bulk density values similar to those observed when the forage systems were established (1.34 kg dm^−3^) and a lower LLWR (0.04 dm^3^ dm^−3^), similar to those found with Ruziziensis grass in monocropping or intercropped. This was probably due to the characteristics of the less vigorous and shallow root system when compared to the treatments containing *B. brizantha* cvs. Xaraes and Paiaguas (Figure 2B), demonstrating less efficiency in soil structural recovery when compared to the other species used. Thus, *B. ruziziensis* has limited potential for physical recovery of the soil, corroborating the results obtained by Flávio Neto et al. [21].

The results obtained by Chioderoli et al. [34], when researching the physical attributes of the soil and the productivity of soybeans in the crop–livestock integration system with forage crops sowing during the maize cover crop season (experimental conditions similar to those evaluated here), were similar in that they found greater biological soil loosening by *B. brizantha* in relation to *B. ruziziensis*. The authors also state that the aggressiveness of the root system of the forage plants provides greater rooting of the crops in succession, especially in the biopores of the grasses inserted into the system. Thus, even if the soil is initially degraded, these species, especially the Xaraes palisadegrass and Paiaguas palisadegrass, improve the soil environment (Figure 2) and enable the subsequent crop, in this case, soybean, to take root. *Brachiaria* species and cultivars can vary in their ability for biological soil loosening [21,22]. The potential superiority of *B. brizantha* has also been quantified by Calonego et al. [35] in a consortium with maize for two consecutive years.

Once the physical degradation of the soil has been diagnosed, grasses of the *Brachiaria* genus, especially those belonging to the *B. brizantha* species, are, in fact, the best strategy for the physical recovery of soils in the Brazilian Cerrado region under the premises of conservation agriculture [11,36] and also an alternative to conventional soil preparation. Currently, in practically all grain-producing regions in Brazil, farmers have been tilling the soil in order to eliminate compaction. The restructuring of the soil by the *B. brizantha* root system makes it more resistant to erosion [22].

In addition, *Brachiaria ruziziensis* has been widely used in conservation agriculture, due to its desiccation efficiency and lower tillage, favoring the plantability of subsequent grain crops [21,22]. On the other hand, this species is less effective at physically restoring the soil when compared to *B. brizantha* cvs. Paiaguas and Xaraes, whether in monocropping or intercropped with maize, whether intended for animal grazing or for the production of the remaining mulch biomass (Figure 2).

### 2.2. Agronomic Aspects for Soybean Crops in Succession

Regardless of the soil cover adopted, the grazing simulation with the cutting of forage plants and the elimination of the remaining mulch biomass of maize from the experimental plots caused a reduction in the biomass yield in relation to free growth (Figure 3). According to Maia et al. [37], although the supply and quality of forage for animals in the off-season when intercropped with maize in the second crop is satisfactory, consumption may compromise soil cover for subsequent agricultural crops, particularly in years when severe water deficits occur, as happened during the grass establishment phase [32]. This is because, in this situation, there is a reduction in biomass production and in the regrowth of forage plants.

The regression coefficients of the decomposition curves of the remaining mulch biomass (Figure 3) show that the free growth forage plants not only produced greater amounts of dry mass, but also showed slower decomposition compared to management with the grazing simulation cut. Figure 3A shows that the preceding maize in monocropping achieved straw productivity of around 6 Mg ha^−1^, similar to Paiaguas palisadegrass intercropped, lower than the biomass produced by the other intercropped forage systems and surpassed by Ruziziensis grass intercropped, with 8.8 Mg ha^−1^, and by the in monocropping systems, with an average productivity of 8.4; 18.4 and 18.8 Mg ha^−1^ for *B. ruziziensis*, *B. brizantha* cv. Paiaguas and Xaraes, respectively. For the systems under the grazing simulation, the production of the remaining mulch biomass varied from 0 Mg ha^−1^ for maize in monocropping (due to the manual removal of all biomass), 2 to 2.8 Mg ha^−1^ for intercropped grass and in monocropping of Paiaguas palisadegrass and 4.0 and 4.7 ruziziensis grass and Xaraes palisadegrass, respectively.

As observed by Paula et al. [5], the consortium between annual crops and *Brachiaria* leads to the suppression of the forage plant’s development, reducing its productive potential. On the other hand, the remaining mulch biomass produced in monocropping forage is higher than that obtained by Dias et al. [23] (5.5 Mg ha^−1^ of Xaraes palisadegrass; 3.1 Mg ha^−1^ of *B. ruziziensis*), even with the adverse weather conditions when the forage plants were planted in our study. It is therefore expected that this system will produce more mulch in years when there is no summer phenomenon.

Regarding the amount of remaining mulch biomass for no tillage in the Brazilian Cerrado region, it should be noted that there is no consensus on an ideal value, ranging from 6 [32] to 10 to 12 Mg ha^−1^ of dry matter [38]. Even so, the biomass production observed, when it is not destined for animal grazing, was higher than adequate for Xaraes palisadegrass and Paiaguas palisadegrass in monocropping and close to these for Ruziziensis grass in monocropping, Xaraes palisadegrass and Ruziziensis grass intercropped, but with greater restrictions for Paiaguas palisadegrass intercropped and biomass from the harvest of maize grown in the second crop (Figure 3A).

The results obtained in the study also show that the possible consumption of forage by animals in the off-season can compromise aspects of conservation agriculture by reducing surface biomass and exposing the soil to the impacts of rain drops. Considerations in this regard highlight the need for specific studies on grazing in crop–livestock integration systems, in contrast to the lower natural protection of the soil from erosion. While forage consumption maximizes animal weight gain on pasture in the livestock phase, it reduces soil cover in the agricultural phase, which can be at least partially offset by greater water infiltration due to biological soil loosening.

The results of mulch production in the system under the grazing simulation cut (Figure 3B), in turn, suggest that in years when forage crops are planted under low rainfall rates, grazing would compromise the production of the remaining mulch biomass for no tillage. In this context, free growth (Figure 3A) could be more advantageous for soybean cultivation, but would exclude the possibility of animal production on pasture.

Alternatively, future research is needed to define management strategies for integrated systems, especially those associated with grazing intensity and pre-drying grass regrowth, in order to establish the relationship between the system’s components and the stability of production (grain, meat and remaining mulch biomass) and the conservation of natural resources. These studies are scarce and inconclusive.

So far, it is undeniable that this intensification of land use, provided by the adoption of crop–livestock integration after the summer crop (the first crop of the agricultural calendar) is sustainable because it is technically feasible and economically resilient because it diversifies production [4,6,39]; socially fair, by keeping human resources in the field, given the optimization and rational use of labor [40]; and environmentally correct [8], by: (i) promoting the sequestration of atmospheric carbon [26], (ii) biologically decompressing the soil, contributing to erosion control and restoring the hydrological cycle [16,21,22], (iii) suppressing diseases and weeds, reducing the use and selection of herbicide-resistant plants [41] and (iv) cycling nutrients, reducing the demand for synthetic fertilizers [23,28]. For these reasons, CLI has recently been used by the Brazilian government to define public policies on agricultural credit, deforestation control and greenhouse gas mitigation. These actions stand out as commitments made under the international climate agreement of Paris—COP 21 [42,43,44].

Figure 3 also shows that all the management systems with grass in free growth had a lower decomposition rate than those with the grazing simulation cut. This is due to the development of the reproductive phase of the forage plants, which leads to an increase in stalks compared to leaves, made up of fibrous materials rich in lignin and cellulose, reducing the rate of the decomposition of the forage plant.

According to Dias et al. [23] the biomass mulch half-life is considered to indicate the durability of the biomass on the soil surface. When adopting integrated systems, it is necessary to choose cover plants that achieve high residue production and keep the soil covered for a longer period, especially in regions such as the Brazilian Cerrado where the decomposition of plant biomass is accelerated by soil and climate conditions [4,23,28].

Figure 3A and Figure 4A show that *B. ruziziensis* in free growth, both intercropped and in monocropping, had a higher biomass decomposition rate (decomposition of 59 and 63% and biomass mulch half-life of 94 and 85 days, respectively) compared to maize and *B. brizantha* grasses (45% average decomposition and half-life ranging from 120 to 150 days). These results corroborate those obtained by Oliveira et al. [23] and are due to the fact that *B. ruziziensis* had a low C/N ratio (<20:1), enhancing decomposition when compared to *Brachiaria brizantha* cultivars, which have higher C/N ratios (>25:1).

For maize, and according to Silva et al. [4], the biomass has a high proportion of stalks, a higher proportion of lignin and, consequently, a very high C/N ratio in the residues, promoting the greater persistence of the soil cover (43:1, as found by the authors) but, nevertheless, promoting little soil cover. In addition, it should be noted that the fact that the evaluation of the decomposition of the remaining mulch biomass began when the soybeans were sowing (172 days after the grain harvest of the previous crop and the deposition of crop residues on the soil) may have contributed to a similar intermediate decomposition (higher than Ruziziensis grass but lower than Xaraes palisadegrass—Figure 4). It is therefore assumed that there was partial decomposition of the remaining mulch biomass from the crop before the start of the evaluation, notably the leaves and unharvested grains, and the residual accumulation of lignified material, consisting mainly of stalks and corms, already at some stage of decomposition.

The differences between the persistence of the remaining mulch biomass of the forage systems subjected to the grazing simulation cut were more discreet and followed the following order: Ruziziensis grass in monocropping (82 days) ≤ Ruziziensis grass intercropped (91 days) = Paiaguas palisadegrass intercropped (99 days) ≤ Xaraes palisadegrass in monocropping (108 days) = Xaraes palisadegrass intercropped (116 days) = Paiaguas palisadegrass in monocropping (116 days). This is because grass regrowth produces more leaves, which are less lignified and therefore more decomposable. Finally, it should be noted that even under a grazing simulation cut regime, the species studied have satisfactory ground cover potential for 120 days (the cycle of the soybean cultivar used), where the longer biomass mulch half-life of the *B. brizantha* cultivars was due to the greater amount of biomass produced and their cespitose morphology.

In addition to providing the aforementioned environmental services, the efficiency of direct sowing in the remaining mulch biomass becomes a deciding factor when choosing the forage plant to be used in conservation systems. In this sense, Figure 5 shows that the plantability and plant population results for the NS 7202 IPRO soybean cultivar did not differ according to the management systems adopted. Figure 5A shows the distribution of plants by the seeder, in which the proportion of plants in normal spacing (0.025 to 0.075 m apart) was 60%, with 23% of double plants (<0.025 m) and 17% of seeding flaws (>0.075 m). According to the performance classification carried out by Correia et al. [45] and Dias et al. [23], the distribution of soybean plants in this study obtained a regular result in terms of seed distribution in the sowing line, considering the gravitational seed distribution mechanism used (between 50 and 70%). The results of the experiment were similar to those obtained by Dias et al. [23] who, evaluating different cultivation systems for no tillage soybeans, obtained an average plant proportion of 60% in normal spacing, 25% of double plants and 18% of flaws in sowing spacing for all treatments. Xu et al. [46] define 50% uniformity (normal spacing) as critical for soybean cultivation and there may be significant yield losses below this.

Research has also reported the low response of soybeans to variations in plant density [47,48]. Figure 5B shows the plant population in the management systems, where the average was 405,036 thousand pl ha^−1^. This attribute is also affected by the cultivar adopted and the capacity for phenotypic plasticity. This is what Carmo et al. [49] found when assessing the agronomic performance of soybeans sowing at different times and with a different spatial distribution of plants.

According to Franchini et al. [50], in crop–livestock integration systems, the timing of desiccation can also significantly influence the plantability and agronomic performance of soybeans in succession. A large amount of biomass under the soil at planting time can increase the skidding rate of the tractor when sowing, as well as causing “wrapping” with the straw accumulated in the seeder row [51]. Due to these problems, an adequate interval is needed between desiccation and sowing, when there is no grazing in the area.

It should be noted that the 31-day interval between desiccation and sowing was sufficient to desiccate all the plants of the species studied and eliminate any interference from any of the grasses evaluated in terms of clogging, tractor/seeder skidding (factors not observed during the planting operation) and planting uniformity, using the direct seeding technology adopted in the experimental conditions. As shown in Figure 5A, there was no difference in the longitudinal distribution of seeds for the treatments or in the plant population (Figure 5B).

According to Brighenti et al. [52], there is variability between the *B. decumbens*, *B. brizantha* and *B. ruziziensis* species in terms of their susceptibility to the herbicide glyphosate, with *B. ruziziensis* being the most susceptible and making it possible to sow with a shorter time interval after desiccation [53]. Therefore, in agronomic planning for conservation agriculture, the process of soil cover formation should be accounted for in the agricultural calendar, depending on the species adopted. For this reason, the results show that any restrictions on the use of *B. brizantha* as a cover crop are not associated with the morphology of this forage plant, but with the physiology of desiccation, particularly in relation to Xaraes palisadegrass, which was less efficient in this process than Paiaguas palisadegrass and ruziziensis grass [54]. Although *B. brizantha* cultivars are considered to be strategic tools for physical soil management [16,21,22], their use is somewhat resisted by farmers due to incorrect desiccation management, commonly associated with the difficulty of desiccation which, in turn, affects the sowing operation of crops in succession, in terms of the loss of the operational efficiency of the machinery.

In areas that use CLI systems together with no tillage, it is recommended that desiccation takes place 15 to 25 days before planting [55]. On the other hand, longer intervals can generally affect the performance of two successive crops due to climatic conditions (concentration of rainfall in the summer and shorter days in the winter), which is why there is resistance to adopting the *B. brizantha* species, which generally requires a longer period. Its adoption as a strategy for recovering the soil’s physical quality must, therefore, be preceded by a diagnosis of soil degradation and also by planning the grass’s management within the agricultural calendar. While on the one, Ruziziensis grass is considered to be easy to desiccate (even though no live or senescing plants were observed at the time of sowing in all the treatments in our study), it has a limited biological soil loosening capacity, as already discussed, and also a lower biomass production capacity for no tillage, also seen in Figure 3.

Considering the aspects of conservation agriculture evaluated here, it is hoped that the development of crops in succession will benefit from the improvement of the soil environment promoted by the prior cultivation of grasses, especially *B. brizantha*. This is because the cultivars of this species stand out both for agricultural production and when intended for animal grazing in relation to their potential for biological soil loosening [21,22] and high forage production in the off-season [37]. However, when evaluating the agronomic attributes of soybeans in succession, this hypothesis was not confirmed. There were no differences between the forage systems adopted, with an average plant height of 61.5 cm, 23.4 axils per plant, 32.3 pods plant^−1^ and a 1000-grain weight of 152.7 g in all treatments. The average grain yield of 4.3 Mg ha^−1^ (Figure 6), also in the forage systems, is above the 2016/2017 Brazilian average, according to Embrapa [56].

Soybean yields were higher than those found by Bonetti et al. [57] when researching the CLI system in free growth management (3.55 Mg ha^−1^) and grazing (3.67 Mg ha^−1^) after brachiaria desiccation. While the climatic conditions during the previous summer crop were characterized by a strong veraison during the grass establishment stage [32], the following one showed satisfactory rainfall distribution, which probably suppressed the effects of the soil structure (Figure 2), decomposition (Figure 3) and persistence (Figure 4) of plant residues on the surface on soybean development, unlike those observed by Dias et al. [23] and Silva et al. [4] in which cropping systems with the presence of forage plants showed the highest yields of soybeans in succession.

In addition, the mechanized sowing of annual crops promotes spatial variation in the soil’s physical properties under no tillage, in which the furrow-opening mechanism causes some decompression in the sowing furrow [18]. This factor, coupled with planting fertilizer also in the sowing furrow and continuous water replenishment due to the occurrence of well-distributed rainfall, may be the reason why the crop has shown productive stability in all the systems evaluated. In addition, Cecagno et al. [58] contributed to a better understanding of soybean performance in crop–livestock integration systems. Evaluating the least limiting water range and soybean yield in no tillage in a long-term experiment in southern Brazil under different grazing intensities (intensive, moderate and no grazing), the authors found that under normal rainfall conditions, the soybean yield depends mainly on the amount and distribution of rainfall and not on the physical quality of the soil.

However, under the experimental conditions evaluated, all the other factors studied showed no direct correlation with the soybean yield (Figure 7). The conservation agriculture indicators associated with soybean yields in this study showed a dispersion of points and a non-significant correlation. The range of grain yields (3.3 to 5.2 Mg ha^−1^), in turn, reflected local control through the experimental blocks.

These results corroborate those observed by Balbinot Júnior et al. [59], in which soybean growth was not limited in soil with a penetration resistance greater than 2.0 MPa (the value used to obtain the LLWR). Marasca et al. [60], when evaluating a Latossolo Vermelho Distroférrico under a no tillage system for soybean cultivation, obtained values ranging from 2.9 to 4.3 MPa which, in turn, did not influence soybean yields.

As can be seen, the soybean crop did not respond to the structural conditions of the soil as a result of the forage systems adopted (no direct correlation between the yield and LLWR, nor the Bd). However, there was also no correlation with the attributes of plantability, plant stand, straw productivity and biomass mulch half-life (Figure 7). In view of the above, it is understood that the attempts at a cause and effect relationship in soybean cultivation may not be well understood, reiterating the statements by Cecagno et al. [58] on the need for evolution in the evaluation of soil science in integrated agricultural production systems, which should consider not only soil parameters, but also physiological indicators in order to represent the perception of the plant in the soil–plant–animal–machine–atmosphere system.

In association, our results demonstrate the potential of species of the *Brachiaria* genus as cover plants for the Cerrado region in conservationist agricultural production systems, especially the Xaraes palisadegrass and Paiaguas palisadegrass, due to their high productivity of remaining mulch biomass, their promotion of biological soil loosening and, furthermore, their ability to achieve this without compromising the population arrangement at the time of the mechanized sowing of successor crops. The greater soil cover provided by these grasses can contribute to the availability of soil water for plants in years when there is a summer drought, as well as reducing soil loss due to erosion as a result of heavy rainfall.

It is hoped that under adverse conditions, such as irregular rainfall and veranicos, the conservation aspects mentioned here will act to mitigate soil compaction, increasing the security and stability of production and, consequently, the sustainability of economic activity. Considering that the adoption of forage systems in the off-season goes beyond phytotechnical aspects, it assumes a prominent role in the development of tropical agriculture and the maintenance of environmental services.

## 3. Materials and Methods

### 3.1. Study Site and Experimental Arrangement

The trial was conducted in an experimental area located at the Instituto Federal Goiano, in the municipality of Rio Verde, state of Goiás, Brazil, with latitude 17°48’34.25” S, longitude 50°54’05.36” W and an altitude of 731 m. The climate is classified, according to Köppen, as Megathermic or Tropical Humid (Aw), subtype Tropical Savannah, with a dry winter and rainy summer. The average annual temperature is 25 °C and the average annual rainfall is approximately 1600 mm. The soil was classified as Latossolo Vermelho Acriférrico típico [19] and Typic Haplustox [61] (USDA, 2014). The physical soil characterization and oxide composition (Table 1) were determined according to Teixeira et al. [62].

The soil was corrected and fertilized according to the results of the chemical analysis of the soil (Table 2), where 30 kg ha^−1^ of N, 250 kg ha^−1^ of P_2_O_5_ and 60 kg ha^−1^ of K_2_O were applied at the time of sowing, according to the recommendations of Souza and Lobato [63], for the soybean cultivation of the Brazilian Cerrado.

To set up the experiment, the following activities were initially carried out: two subsoiling operations crossed at a depth of 0.40 m, one plowing and two harrowing operations, with the aim of eliminating the historical tension in the area. Subsequently, the soil was compacted using an agricultural tractor with a tare weight of 4.5 Mg when the soil was close to field capacity and passed four times in the same place throughout the area, following the procedure described by Guimarães Júnnyor et al. [64].

The treatments were implemented by Torino et al. [32] in January 2016, corresponding to first crop. Our study corresponds to second crop (when the seasons change from rainy summer to dry winter) and was carried out in a randomized complete block design with four replications in subdivided plots. In the plots, 14.0 m long and 6.5 m wide, seven treatments were evaluated: *Brachiaria brizantha* cv. BRS Paiaguas, *Brachiaria brizantha* cv. Xaraes and *Brachiaria ruziziensis*, in monocropping and intercropped with maize, as well as maize in monocropping (control/additional). In the subplots, there were two grass management systems: free growth and grazing simulation cutting, with the aim of quantifying the potential for structuring the soil by the root system and the grass cutting management (considering the possibility that when cut/grazed, there will be a high physiological response of regrowth and rooting), without the influence of animal trampling, which could cause additional compaction [16].

The treatments represent the variants of the crop–livestock integration system adopted in Brazil. The maize–grass consortium with forage cut represents the animal grazing after the grain crop and monocropping forage with grazing simulation are designed to replace the second grain crop. The purpose of growing grass in free growth is to produce biomass for subsequent no tillage. Maize in monocropping, on the other hand, had the remaining mulch biomass removed from the plot, in simulation of consumption by grazing animals.

On 26 January 2016, the maize was sowing mechanically and the forage plants were oversown in the in monocropping systems. At 22 days after sowing (DAS), the forage species were oversown in the consortium plots, along with the top dressing of maize. It should be noted that during the ear formation stage, there was a strong veranico, totaling 44 days without rain [32]. The maize was harvested at 126 DAS and the forage in the in monocropping system had four cuts carried out on 31 March, 3 June, 4 August and 10 October 2016. In the intercropping systems, the first cut took place after June 03, after the maize crop harvest, and there was a total of three cuts.

### 3.2. Soil Sampling and Physical Analysis

In October 2016, soil with preserved structure was sampled after the grass cycle, in all subplots and in three soil layers (0–0.05; 0.05–0.10 and 0.10–0.20 m), using volumetric rings with the aid of a Uhland sampler, to assess the potential for biological soil loosening, according to Flávio Neto et al. [21]. Deformed samples were also collected in the layers described, used to determine the permanent wilting point (matric potential of −1.5 MPa) using the Richards Extractor [62].

The samples were slowly saturated and then subjected to a matric potential of −0.006 MPa in porous plate funnels and the soil water content (θ) obtained was considered equivalent to the field capacity of the soil [12]. Each sample was then adjusted to two different soil water contents, ranging from 0.05 (the lowest content found in the field during the dry season of the year) to 0.36 dm^3^ dm^−3^ (the highest content found at −0.006 MPa) and submitted to the penetrometry test. This test used a MARCONI-MA 933/30 benchtop penetrometer (Piracicaba, Brazil) equipped with an electronic speed variator (10 mm min^−1^) and a data recording system, obtaining the penetration resistance (PR) for each volumetric soil water content (θ), according to the method used by Severiano et al. [29]. Finally, the samples were dried in an oven at 105 °C for 48 h to determine bulk density (Bd). Total porosity (TP) was calculated using Equation (1), according to Blake and Hartge [65], based on Bd and particle density (Pd) considered to be 2.74 g cm^−3^.
(1)$$\mathrm{TP}=\left[1-\left(\mathrm{Bd}/\mathrm{Pd}\right)\right]$$

Soil penetration resistance curve (SPRC) was obtained by modeling PR as a function of θ and Bd, following the non-linear model proposed by Busscher [66] (Equation (2)):(2)$$\mathrm{PR}=0{.01\theta }^{-0.96}{\mathrm{Bd}}^{6.51};\;R^{2}=0.83\;(p\;<\;0.01)$$

The least limiting water range (LLWR) was determined according to the procedures described in Silva et al. [67]. The upper limit was considered to be the lowest soil water content between that retained in the field capacity (θFC) [29] or that in which the soil water content guarantees a minimum air porosity (θAP) of 10% [68], calculated for each sample using Equation (3):(3)$$\theta_{\mathrm{AP}}=\mathrm{TP}-0.1$$

The lower limits were the soil water content retained at the matric potential of −1.5 MPa with the soil water content at the permanent wilting point (θPWP) [69] and/or the soil water content corresponding to the soil penetration resistance of 2.0 MPa (θPR), obtained using Equation (2).

### 3.3. Sampling and Decomposition of Straw

After sampling the soil, the grasses were desiccated 31 days before sowing the soybeans, spraying the herbicide Glyphosate (3.5 L ha^−1^) over the entire area. At 14 days after desiccation, when the grass showed visual symptoms of senescence, samples of the straw were collected from 2.0 m^2^ in each sub-plot, with the exception of the maize in monocropping sub-plot intended for animal grazing, in which the straw remaining from the grain crop was previously eliminated in a grazing simulation. In this way, this treatment remained uncovered at the time of sowing and throughout the soybean cycle.

The material was weighed and part was separated to determine the dry matter mass in an oven (55 °C for 72 h). Based on these values, the mulch biomass yield per hectare of each treatment was determined. The rest of the material was proportionally divided into four nylon bags (0.3 m × 0.3 m), based on the weight corresponding to the initial material obtained in the 2.0 m^2^ evaluated.

Each nylon bag represented a decomposition time in the field, corresponding to 30, 60, 90 and 120 days after senescence, considering the initial weight as 0 days. At the end of this period, the bags from each time period were collected and the material was washed in distilled water and dried in a forced circulation oven (55 °C until constant weight) to determine the mass of dry matter remaining, based on the difference between the initial and final weights.

To describe the straw decomposition curve, the remaining mulch biomass data, as a function of decomposition time (0 to 120 days), were adjusted to the exponential mathematical model described by Wieder and Lang [70] (Equation (4)):(4)$$P=P_{0}e^{\left(-kt\right)}$$
where *P* is the amount of dry biomass existing at time *t*, in days; *P_0_* is the fraction of potentially decomposable straw; and *k* is the biomass decomposition constant.

The biomass mulch half-life (*t*_1/2_), in other words, the time needed for 50% of the remaining biomass to be decomposed, was calculated using Equation (5), according to Paul and Clark [71]:(5)$$t_{1/2}=0.69/k$$

Soybeans were sowed on 18 November 2016 using the NS 7202 IPRO cultivar (NIDERA, São Paulo, Brazil). This cultivar has high productivity, favorable architecture for disease control, adaptation to planting times and indeterminate growth. The cultivar’s seeds were inoculated with *Bradyrhizobium japonicum* at a ratio of 1 kg of inoculant to 50 kg of seeds. A concrete mixer was used to homogenize the inoculant, graphite, fungicide (Cropstar^®^, Yarmouth, IA, USA) and insecticide (Cruiser^®^), specific for seed treatment. Fertilization was carried out in the planting furrow, according to the soil analysis (Table 1 and Table 2).

### 3.4. Evaluation of Plantability, Plant Health Treatments and Soybean Productivity

The soybean crop was planted using a 6600 tractor traction (John Deere, Moline, IL, USA): 4 × 2 TDA with 121 hp and a Massey Ferguson no tillage seeder/adubator, model MF 500 Series L-M-SEED, with nine rows spaced at 0.5 m and with the furrow opening for depositing the seed/adube using a double offset disc system. The seeder was set to a population of 400,000 plants per hectare, which is what is recommended for the cultivar in central-west Goiás [72], with a seeding depth of 5 cm, as recommended by the cultivar. An average speed of 4.8 km h^−1^ was used, as recommended by Rauta [73].

Pest control began at 30 days after emergence (DAE), with granulated bait (pyrazole chemical composition fipronil 0.01%) for ant control and the fungicides trifloxystrobin + cyproconazole (300 mL ha^−1^) preventatively for Asian rust (*Phakopsora pachyrhizi*) at 60 DAE. Pests were surveyed at 72 DAE (*Colaspis* sp., *Bemisia tabaci*, *Megascelis* sp, *Spodoptera albula* and *Euschistus heros*). They were then controlled with the insecticide Thiamethoxam + Lambda-cyhalothrin (200 mL ha^−1^).

The plantability assessment was carried out 15 days after sowing (DAS) with the aid of tape measures on five central sowing lines, over a length of 2 m, which totaled 4.5 m^2^ per sub-plot. During the evaluation, the following were observed: “double spacing” (D)—0.5 times smaller than the average spacing established; “acceptable or normal spacing” (A)—0.5 to 1.5 times the average spacing established; and “seeding flaws” (F)—1.5 times larger than the average spacing established, according to Associação Brasileira de Normas Técnicas [74], Dias et al. [23]. The average spacing established for the spacing and population adopted was 0.05 m between plants.

To determine soybean yield, all the plants in the 4.5 m^2^ area used to assess plantability were harvested at 143 DAS, 10 of which were separated for agronomic assessment of the crop, based on the following characteristics: plant height, considering the distance between the soil surface and the apical end of the main stem of the plant; number of pods, obtained by counting; and number of axils with and without pods, counted on the same 10 plants collected previously.

The plants were then threshed and the mass was determined. The values obtained were transformed into 1000-grain weight and yield (Mg ha^−1^), corrected to 13% humidity. Rainfall and temperature were monitored daily throughout the soybean cycle (Figure 8).

### 3.5. Statistical Analysis

The results of the physical attributes of the soil, biomass mulch half-life, plantability and soybean development were subjected to analysis of variance and the means were compared using the Tukey test (*p* < 0.05) for the qualitative variables and regression for the quantitative variables, when significant. In addition, Pearson’s correlation coefficient (r) was used to assess the correspondence between all the attributes relating to aspects of conservation agriculture and the development of the soybean crop in integrated agricultural production systems.

## 4. Conclusions

The cultivars of *B. brizantha* cv. Xaraes and BRS Paiaguas are promising alternatives for conservation agriculture because they provide the environmental services of biological soil loosening and have a high potential for producing remaining mulch biomass for subsequent crops. Therefore, it is possible to keep the soil cover by biomass mulch throughout the soybean in succession.

The plantability of soybeans in the biomass of *Brachiaria* sp. grasses is not affected and the recommended plant population is obtained, as long as the time needed for the senescence and formation of the sowing bed is respected.

Under the study conditions, the aspects of conservation agriculture evaluated did not affect the agronomic development and productivity of soybeans, probably due to the regular distribution of rainfall and also the characteristics of the cultivar.

## Figures and Tables

**Figure 1 plants-12-03746-f001:**
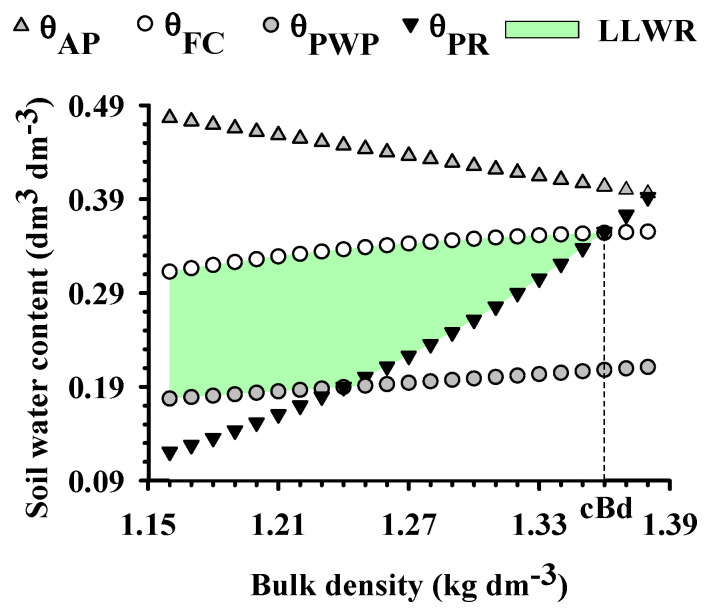
Variation of soil water content (θ) with the increase in bulk density (Bd), at the critical limits of field capacity (θ_FC_, −0.006 MPa), permanent wilting point (θ_PWP_, −1.5 MPa), air porosity at 10% (θ_AP_) and soil penetration resistance of 2.0 MPa (θ_PR_) of Latossolo Vermelho Acriférrico cultivated in crop–livestock integration systems, in the municipality of Rio Verde, Goiás, Brazil. The hatched area represents the least limiting water range (LLWR); cBd: critical density for plant development.

**Figure 2 plants-12-03746-f002:**
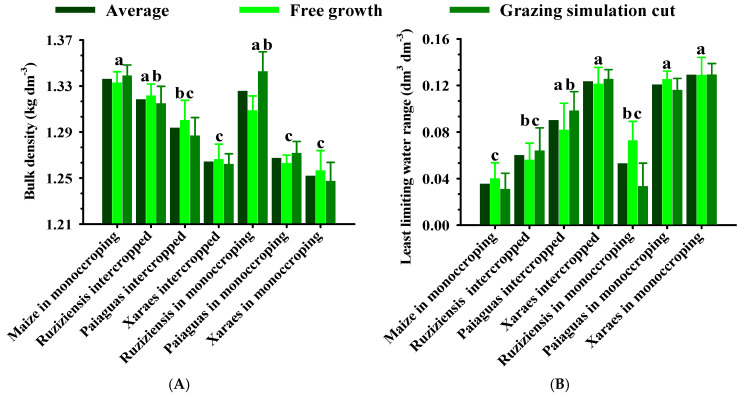
Soil physical quality attributes [bulk density (**A**) and changes in the least limiting water range (**B**)] of Latossolo Vermelho Acriférrico in conservation agriculture systems in the municipality of Rio Verde, Goiás, Brazil. Averages between forage systems followed by the same lowercase letter do not differ by the Tukey test (*p* < 0.05). Vertical bars indicate the standard error of the mean.

**Figure 3 plants-12-03746-f003:**
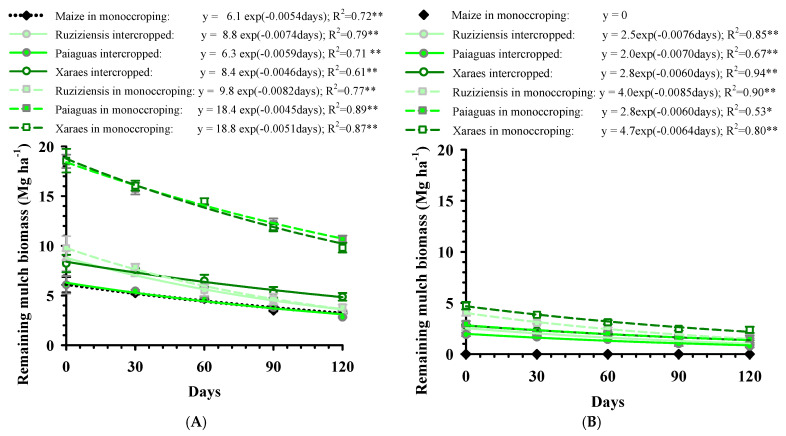
Decomposition curve of remaining mulch biomass of *Brachiaria* and maize in free growth (**A**) and under grazing simulation cut (**B**) and cultivated in crop–livestock integration systems under a Latossolo Vermelho Acriférrico in the municipality of Rio Verde, Goiás, Brazil. Vertical bars indicate the standard error of the mean. *: (*p* < 0.05); **: (*p* < 0.01).

**Figure 4 plants-12-03746-f004:**
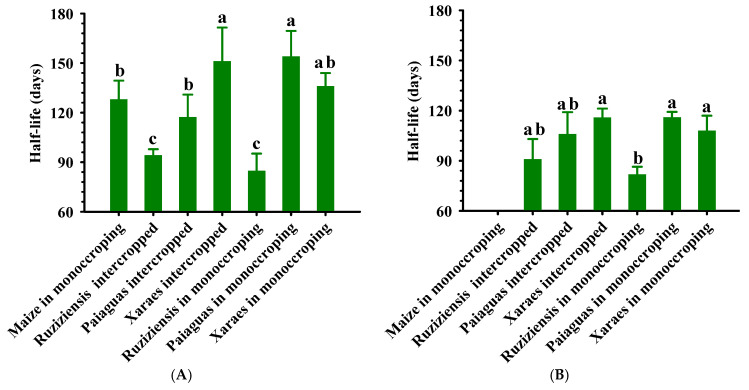
Biomass mulch half-life of *Brachiaria* and maize, in free growth (**A**) and under grazing simulation cut (**B**) and cultivated in crop–livestock integration systems under an Latossolo Vermelho Acriférrico in the municipality of Rio Verde, Goiás, Brazil. Averages between forage systems followed by the same lowercase letter do not differ by the Tukey test (*p* < 0.05). Vertical bars indicate the standard error of the mean.

**Figure 5 plants-12-03746-f005:**
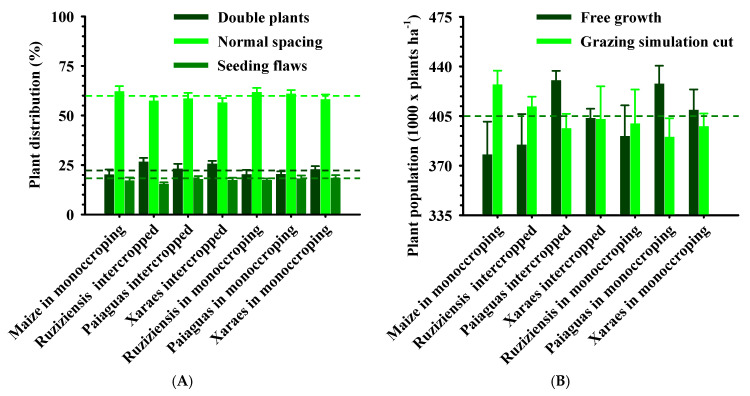
Soybean plantability as a result of conservation agriculture production systems in terms of longitudinal plant distribution (**A**) and plant population (**B**) on Latossolo Vermelho Acriférrico in the municipality of Rio Verde, Goiás, Brazil. Dashed lines indicate the mean and vertical bars the standard error of the mean.

**Figure 6 plants-12-03746-f006:**
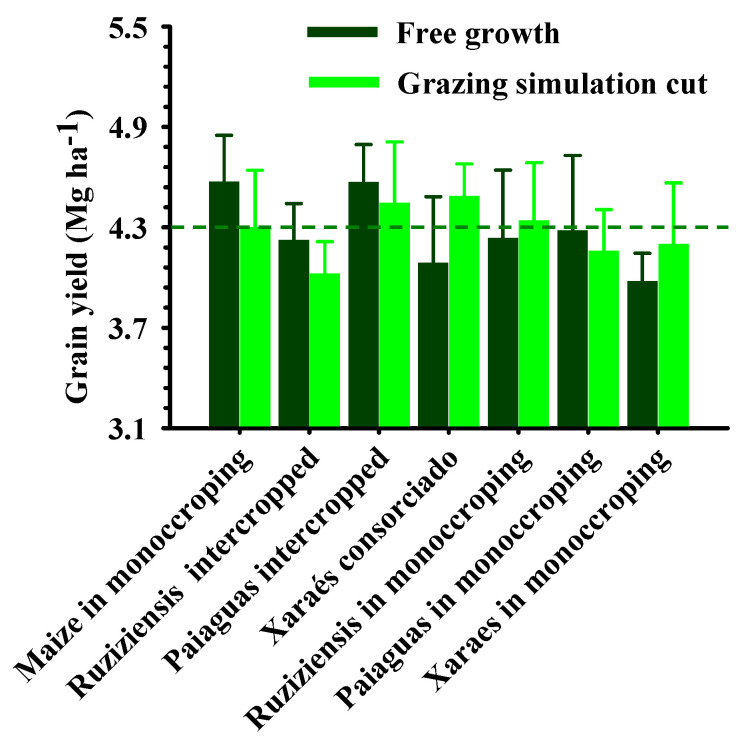
Productivity of soybeans grown in different crop–livestock integration systems under an Latossolo Vermelho Acriférrico in the municipality of Rio Verde, Goiás, Brazil. Dashed lines indicate the mean and vertical bars the standard error of the mean.

**Figure 7 plants-12-03746-f007:**
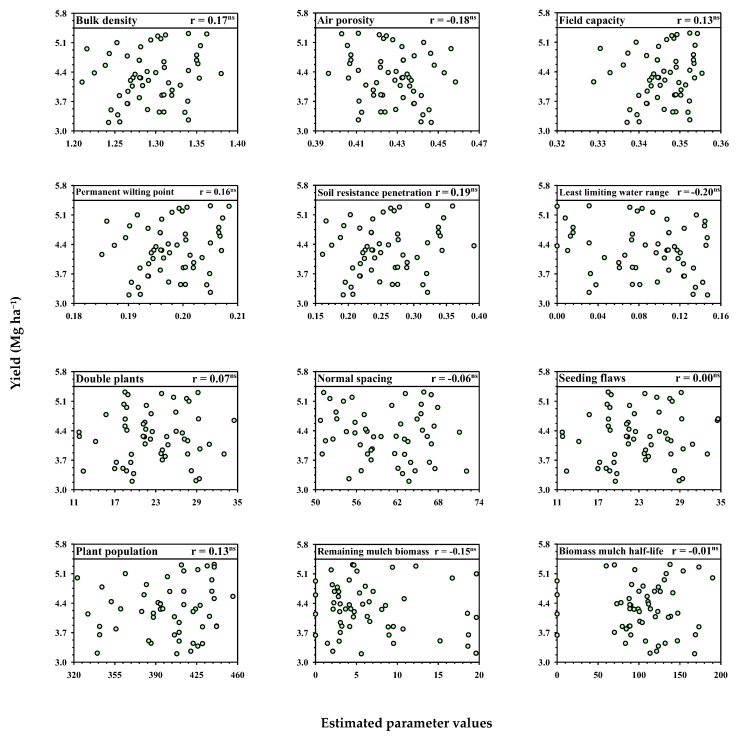
Pearson’s correlation between conservation agriculture indicators and soybean productivity in an Latossolo Vermelho Acriférrico in the municipality of Rio Verde, state of Goiás, Brazil, under different forage and soil management systems. ns: not significant.

**Figure 8 plants-12-03746-f008:**
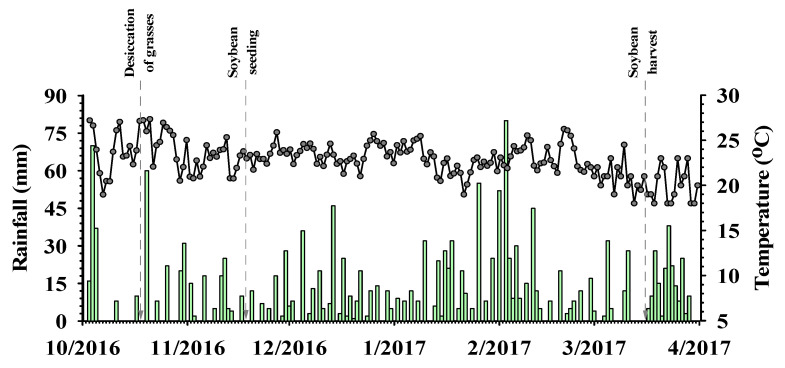
Rainfall (mm) and temperature (°C) in the municipality of Rio Verde, Goiás, Brazil, during the course of the experiment. Bars indicate rainfall and dots indicate temperature.

**Table 1 plants-12-03746-t001:** Physical and chemical characterization of Latossolo Vermelho Acriférrico in the municipality of Rio Verde, state of Goiás, Brazil.

Layer	Pd ^(1)^	Bd ^(2)^	Texture ^(2)^			Oxides in Sulfuric Attack			Ki	Kr
			Sand	Silt	Clay	SiO_2_	Al_2_O_3_	Fe_2_O_3_		
(m)	(kg dm^−3^)		--------------------------- (g kg^−1^) ---------------------							
0–0.20	2.80	1.34	350	200	450	46	198	224	0.39	0.23

^(1)^ Pd: Particle density; ^(2)^ Bd: Bulk density; Ki: molecular relation (SiO_2_/Al_2_O_3_); Kr: molecular relation SiO_2_: (Al_2_O_3_ + Fe_2_O_3_). Attributes determined according to Teixeira et al. [62].

**Table 2 plants-12-03746-t002:** Assortment complex in the 0–0.20 m layer of typical Latossolo Vermelho Acriférrico in the municipality of Rio Verde, Goiás, Brazil before the experiment was set up.

Ca	Mg	Al	Al + H	P	K	S	Zn	B	Cu	Mn	Mo	V ^(1)^	*m* ^(2)^	O.M. ^(3)^	pH
----- cmol_c_ dm^−3^ -----				--------------- mg dm^−3^ --------------								--- % ---		g kg^−1^	
1.83	0.75	0.01	5.15	2.8	34	10.55	0.62	0.17	3.82	25.95	25.71	34.25	0.37	25.17	4.96

^(1)^ V: base saturation; ^(2)^
*m*: aluminum saturation; ^(3)^: O.M.: organic matter. P: Determined by the extractor Mehlich. pH at CaCl_2_. Attributes determined according to Teixeira et al. [62].

## Data Availability

The data presented in this study are available upon request from the corresponding author.

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
