# Peer review of "Crop-Livestock Integration Improves Physical Soil, Agronomic and Environmental Aspects in Soybean Cultivation"

_plants, 2023, doi:10.3390/plants12213746_

Round 1
Reviewer 1 Report
Dear Authors,
Congratulation for the good work.
1. Abstract is good and to the point.
2. Introduction is written very well.
3. Material and Methods section has described in detailed
4. Results and Discussion are appropriately discussed
5. In my opinion Conclusion section should be fleshed out.
Author Response
Por favor, verifique o anexo.

Reviewer 2 Report
Ln-24: The description of the results is too brief and should specify which physical properties of the soil have been changed.
Ln-36: It is more appropriate to replace 'tool' with 'technology' here.
Ln-64: Consider whether the use of causal relationships here is correct.
Ln-111: It is recommended to add the significance analysis of decomposition rate between different forage systems with the same decomposition day and the same forage system at different decomposition days.
Ln-115: Lack of explanation for letter annotations in the figure.
Ln-130: Lack of specific significant differences in LLWR values among different feed systems.
Ln-137: Delete this sentence or put it before the LLWR value analysis.
Ln-469: References should be added here.
Ln-489: Should 'cut' be replaced with 'cutting'?
Ln-490: Since we want to eliminate the impact of animal trampling, why do we still use grazing simulation cutting?
Ln-506: Lack of introduction to forage harvesting time.
Overall good, but pay attention to the use of causal relationships and refine the sentence.
Reviewer 3 Report
Line 77 - The first time, insert the name in full into the text and the acronym in brackets
Line 280 C/N. Always use the same symbols. Either C/N or C:N
Lines 481-500 -The description of the entire experimental setup (number/type of treatments, plot, subplot, crop rotation) is confusing and difficult to interpret. It would be appropriate for the authors to improve the description by also using shorter sentences and introducing at least a couple of schemes, one on an experimental level and one on crop rotation with the operations carried out
Line 601 - Figure 8 should be renumbered as figure 1. Consequently, all the figures in the captions under the figures and in the text should be renumbered
